# Episiotomies and obstetric anal sphincter injuries following a restrictive episiotomy policy in France: An analysis of the 2010, 2016, and 2021 National Perinatal Surveys

Raphaele Houlbracq[1], Camille Le Ray[1,2], Béatrice Blondel[1], Nathalie Lelong[1], Anne Alice Chantry[1,3☯], Thomas Desplanches[1,4,5☯]*, ENP2021 Study Group[¶]

1 Université Paris Cité, Center of Research in Epidemiology and StatisticS/CRESS/Obstetrical Perinatal and Pediatric Epidemiology Research Team (EPOPé), INSERM, INRAE, Paris, France, 2 Port Royal Maternity Unit, Cochin Hospital, Assistance Publique-Hôpitaux de Paris, Université Paris Cité, Paris, France, 3 Midwifery Universitary Department, Université Paris Cité, Paris, France, 4 Geneva School of Health Sciences, HES-SO University of Applied Sciences and Arts Western Switzerland, Geneva, Switzerland, 5 CHRU Dijon, Department of Gynecology, Obstetrics, Fetal Medicine and Infertility, Dijon, France

☯ These authors contributed equally to this work.
¶ Collaborators of the ENP2021 Study Group: Camille Le Ray (Inserm EPOPé), Nathalie Lelong (Inserm EPOPé), Hélène Cinelli (Inserm EPOPé), Béatrice Blondel (Inserm EPOPé), Nolwenn Regnault (Santé publique France), Virginie Demiguel (Santé publique France), Elodie Lebreton (Santé publique France), Benoit Salanave (Santé publique France), Jeanne Fresson (Direction de la Recherche, des Etudes, de l'Evaluation et des Statistiques), Annick Vilain (Direction de la Recherche, des Etudes, de l'Evaluation et des Statistiques), Thomas Deroyon (Direction de la Recherche, des Etudes, de l'Evaluation et des Statistiques), Philippe Raynaud (Direction de la Recherche, des Etudes, de l'Evaluation et des Statistiques), Sylvie Rey (Direction de la Recherche, des Etudes, de l'Evaluation et des Statistiques), Khadoudja Chemlal (Direction Générale de la Santé), Nathalie Rabier-Thoreau (Direction Générale de la Santé), Frédérique Colombet-Migeon (Direction Générale de l'Offre de Soin)
* thomas.desplanches@hesge.ch

**Data Availability Statement:** Data cannot be shared publicly because they are protected by

## Abstract

### Background

The French guidelines have recommended a restrictive policy of episiotomy since 2005. We aimed to assess variations in the prevalence of both episiotomy and obstetric anal sphincter injury (OASI) from the 2010, 2016, and 2021 National Perinatal Surveys.

### Methods and findings

A total of 29,750 women who had given birth to a live infant by vaginal delivery were included. For instance, in 2021, 22.3% of women were over 35 years old, 17.7% were born outside of France, 11.3% had a body mass index (BMI) of 30 kg/m$^2$ or higher, and 39.9% were primiparous. Episiotomy and OASI (third- and fourth-degree tears) were identified from medical records. We described the overall prevalence of outcomes, and then by obstetrical clinical contexts using a seven-group obstetric classification of women. Variations between 2010 (reference), 2016, and 2021 were analyzed by Cochran–Armitage tests and using Poisson regression models adjusted for maternal age, BMI, country of birth, antenatal classes, suspicion of fetal macrosomia, and neuroaxial analgesia during labor, the

French law as part of a mandatory national audit. Anonymized data are available for researchers who meet the criteria for access to confidential data. Requests for data access should be addressed to the Obstetrical, Perinatal and Pediatric Epidemiology Research Team using the following email: epope.U1153@inserm.fr.

**Funding:** The French national perinatal surveys (ENP) were supported by the Ministry of Health (Direction de la Recherche, des Etudes, de l'Evaluation et des Statistiques (DREES), Direction Generale de la Sante (DGS), and the Direction Generale de l'Organisation des Soins (DGOS)), and, in 2010, by the Fonds d'Intervention en Sante publique, and in 2016 and 2021 by Sante publique France. The funders had no role in study design, data collection and analysis, decision to publish, or preparation of the manuscript.

**Competing interests:** The authors have declared that no competing interests exist.

**Abbreviations:** aRR, adjusted risk ratio; BMI, body mass index; CI, confidence interval; CNIL, Commission Nationale de l'Informatique et des Libertés; ENP, Enquête Nationale Périnatale; OASI, obstetric anal sphincter injury.

professional who attended the birth, the annual number of deliveries, and the maternity unit status to account for changes in women's characteristics and obstetric practices.

The overall prevalence of episiotomy decreased significantly from 25.8% (95% confidence interval (CI) 25.0 to 26.7) in 2010, to 20.1% (95% CI 19.3 to 20.9) in 2016, and 8.3% (95% CI 7.8 to 8.9) in 2021 (adjusted risk ratio (aRR) 0.33, 95% CI 0.30 to 0.35). This reduction was observed in all groups of the classification (Cochran–Armitage tests $P < 0.001$), ranging from −33.0% in Group 2a [nulliparous term cephalic singleton with forceps delivery] to −94.0% in Group 7 [multiple pregnancy]. The difference in overall prevalence of OASI between 2010 (0.7%) and 2021 (1.0%) was not statistically significant after adjustment (aRR 1.24, 95% CI 0.91 to 1.68). By groups of classification, the prevalence of OASI increased significantly only in Group 2b [nulliparous term cephalic singleton with spatula delivery] from 2.6% (95% CI 1.2 to 5.6) in 2010 to 9.6% (95% CI 6.2 to 14.7) in 2021 (aRR 3.69, 95% CI 1.50 to 9.09), and did not differ statistically significantly in Group 2a [nulliparous term cephalic singleton with forceps delivery] from 3.2% (95% CI 1.8 to 5.7) in 2010 to 5.7% (95% CI 3.4 to 9.5) in 2021 (aRR 1.78, 95% CI 0.81 to 3.90).

The main limitations of this study include the failure to take into account some potential confounding factors and the inability to analyze some groups of the studied population (8.5% of the sample) because of the very small number of events in these groups.

## Conclusions

The significant overall reduction in the prevalence of episiotomy in France has not been followed by an overall increase in OASI. However, subgroup analyses revealed a significant rise in OASI among nulliparous women giving birth by spatula (Group 2b), and a clinically relevant but statistically nonsignificant rise among nulliparous women delivering by forceps (Group 2a), while the prevalence of episiotomy significantly decreased. These results should be interpreted with caution given the low prevalence of OASI in some subgroups. Further research is needed to predict the optimal rate of episiotomy for instrumental deliveries. In hospitals with high episiotomy rates, our findings suggest that episiotomy could be safely reduced for spontaneous vaginal deliveries to comply with international guidelines and women's requests.

## Author summary

### Why was this study done?

- Episiotomy is a common surgical procedure during childbirth intended to facilitate birth and can be used by clinicians to prevent obstetric anal sphincter injuries.

- Obstetric anal sphincter injuries are a rare but severe complication of vaginal delivery, impacting women's short-term and long-term health and well-being.

- International guidelines have recommended the restrictive use of episiotomy for over 15 years, but its prevalence varies significantly by country and clinical context.

- It was important to evaluate the restrictive episiotomy policy that has been endorsed for more than 20 years to ensure that it has not led to an increase in obstetric anal sphincter injuries.

### What did the researchers do and find?

- We analyzed the medical records of 29 750 women who had vaginal delivery to identify episiotomy and obstetric anal sphincter injuries, using data from the 2010, 2016, and 2021 French National Perinatal Surveys.

- We described the prevalence of these outcomes overall and then by obstetrical clinical contexts using a seven-group obstetric classification specifically designed to assess episiotomy and obstetric anal sphincter injuries (e.g., in nulliparous with a term cephalic singleton and a forceps delivery, or in multiparous with a term cephalic singleton and a spontaneous delivery).

- Variations between 2010, 2016, and 2021 were analyzed with models taking changes in maternal characteristics and obstetric practices over time into account.

- Our results highlight the significant reduction in the overall prevalence of episiotomy without a corresponding overall increase in obstetric anal sphincter injuries. However, women in specific groups (nulliparous with term cephalic singleton and a forceps or a spatula delivery) experienced a 2- and even 3-fold increase in obstetric anal sphincter injuries.

### What do these findings mean?

- The implementation of a restrictive episiotomy policy can effectively lead to a steep reduction in the episiotomy rate (less than 10% overall), minimizing the number of unnecessary episiotomies and complying with women's requests.

- The implementation of a restrictive episiotomy policy leading to a steep reduction in the episiotomy rate is not necessarily followed by an increase in obstetric anal sphincter injuries.

- Our results and recent literature suggest that there is a need to reconsider indications of restrictive episiotomy policies for instrumental deliveries in nulliparous women.

- Our results should be interpreted with caution due to the low number of obstetric anal sphincter injuries in some subgroups. Future studies with a high level of evidence should be carried out to predict the optimal rate of episiotomy for nulliparous women requiring instrumental delivery.

### Introduction

Episiotomy is one of the most common surgical procedures performed during childbirth. While this incision of the perineum is intended to facilitate birth and to prevent potentially severe obstetric injury to the anal sphincter (OASI), it is also associated with severe maternal complications such as postpartum hemorrhage [1,2], urinary retention [3], infection [4,5], dyspareunia [6], anxiety [7], and post-traumatic stress disorder [8].

Evidence from the literature therefore suggests the benefit of a restrictive practice of episiotomy [9–11] and international guidelines have been consistent for many years in recommending that episiotomy use should be limited during spontaneous vaginal birth [12–16]. However, some studies have shown that the implementation of a restrictive episiotomy policy in the overall population may expose women to higher rates of OASI [17–19]. The decision not to perform an episiotomy may be more difficult in certain cases, such as in instrumental vaginal delivery, where the use of episiotomy for the prevention of OASI is still debated [15,16]. Considering the ongoing challenge of targeting women for whom an episiotomy could be beneficial, a seven-group classification system was proposed in 2019 to provide a clinically relevant framework for assessing episiotomy practices, taking into account the obstetric context [20].

In response to high rates of episiotomies in France compared to international data [9], (71% in nulliparous women in 2003) [21], a French representative association of service users in perinatal care convinced authorities and the French National College of Gynecologists and Obstetricians (CNGOF) to draw up recommendations for clinical practice on episiotomy in 2004. One year later, the CNGOF published guidelines recommending a restrictive episiotomy policy aiming for less than 30% of deliveries with a mediolateral episiotomy [12]. This restrictive episiotomy policy has been reiterated in all sets of guidelines associated with vaginal delivery, until the revision of the guideline for the perineal prevention and protection in obstetrics published in 2018 [12,16,22–25].

The French national population-based perinatal surveys, which aim to monitor perinatal health indicators in France, provide an opportunity to investigate how this restrictive policy has been implemented in France over a decade. The aim of our study was to assess changes in the overall prevalence of episiotomy and OASI between 2010 and 2021, using a clinically relevant classification of obstetric conditions, to better understand variations in the episiotomy rate depending on clinical context.

## Methods

In France, 99% of women give birth in maternity units [26]. In public maternity units (71% of births in 2021) [27], midwives perform all non-instrumental vaginal deliveries and episiotomies if needed. Instrumental vaginal deliveries are performed by obstetricians who perform episiotomies as well if needed in these cases. In private maternity units, obstetricians carry out all childbirth procedures, including non-instrumental and instrumental vaginal deliveries and episiotomies. Since 2005, French guidelines have recommended mediolateral episiotomy [12]. In 2010, 97.5% of episiotomies were mediolateral and 2.5% were median [28].

### Data sources and ethical approval

The National Perinatal Surveys [Enquête Nationale Périnatale—ENP] are regular nationwide population-based cross-sectional surveys including all births (live births and stillbirths) after 21 weeks of gestation or birthweight at least 500 g during a 1-week period in all maternity units in France [27]. In each edition, data is collected from 3 different sources. First, during the postpartum stay, face-to-face interviews of mothers are conducted by midwives according to a standardized questionnaire to collect data on individual and pregnancy characteristics. Second, characteristics related to maternal health and obstetric care are collected from medical records. Third, the head of each maternity unit completes a specific questionnaire regarding the organization of the unit.

For each edition of the ENP, women were individually asked orally for their consent to participate, and they could agree, or fully or partially decline to take part in the survey. Authorizations were obtained to conduct the successive ENPs and secondary analyses from the

following organizations: the Label Committee (Label of general interest and statistical quality, Visa n˚2021 × 701SA, Comité National de l'Information Statistique (CNIS) visa n˚ 2016X703SA, and CNIS visa n˚ 2010X716SA), the local independent ethics committee, the Committee of Ethics and Scientists for Research, and the National Commission on Informatics and Liberty (Commission Nationale de l'Informatique et des Libertés–CNIL) (CNIL-2010- n˚ 909003, CNIL-2016- n˚ 915197, and DR-2020-391) [29].

## Population

Our study population, obtained from the 2010, 2016, and 2021 ENPs, included all women who had a vaginal delivery of a live infant in all the maternity hospitals of mainland France. Women who refused access to their medical records were excluded, corresponding to 0% in 2010, 0.1% in 2016, and 1% in 2021. Women in overseas territories were excluded because of particularities in their characteristics and in the organization of care in these areas. We excluded women younger than 18 years seeing as they were not included in the 2016 survey. In addition, we excluded women with missing data for the main and secondary outcomes or missing data for characteristics required to apply the classification of obstetric conditions.

## Outcomes

Our main outcome was episiotomy, analyzed as a dichotomized variable. The secondary outcome was OASI (i.e., third-degree and fourth-degree tears), defined according to the Royal College of Obstetricians and Gynaecologists classification and the French guidelines [14,16]. This information was obtained from women's medical records.

## Studied variables

We considered the following maternal characteristics: maternal age, maternal country of birth, maternal socioeconomic status, and body mass index (BMI) before pregnancy (kg/m$^2$). Studied characteristics before and during delivery were suspicion of fetal macrosomia, analgesia, and birthweight. Finally, the organizational characteristics included the qualification of the health care provider present at delivery for spontaneous birth (obstetrician or midwife), the annual number of deliveries (<1,500, 1,500–3,499, ≥3500 deliveries per year), and the status of the maternity unit (public university hospital, other public hospital, private hospital).

## Seven-group classification for evaluating episiotomy practices

As described in Desplanches and colleagues, studying episiotomy and OASI needs to take into account specific clinical contexts [20]. These have been synthetized in a seven-group classification system based on the following items: number of fetuses, gestational age at birth (in completed weeks' gestation), fetal presentation, parity, and mode of delivery [20]. The classification assigns all women with a vaginal birth to one of the 7 independent and mutually exclusive groups (Box 1). While forceps and spatula were grouped together in this classification, we assessed them separately to better study episiotomy and OASI rates for each instrument. Consequently, we detailed the type of instruments used, i.e., forceps, spatula, or vacuum, in groups 2 (2a, 2b, 2c, respectively) and 4 (4a, 4b, 4c, respectively).

## Box 1: Seven-group classification according to Desplanches and colleagues [20]

| Groups | Seven-group classification |
|---|---|
| 1 | Nulliparous women with a single cephalic pregnancy at ≥37 weeks of amenorrhea, non-instrumental delivery |
| 2 | Nulliparous women with a single cephalic pregnancy at ≥37 weeks of amenorrhea, instrumental delivery |
| | 2a forceps delivery |
| | 2b spatula delivery |
| | 2c vacuum delivery |
| 3 | Multiparous women with a single cephalic pregnancy at ≥37 weeks of amenorrhea, non-instrumental delivery |
| 4 | Multiparous women with a single cephalic pregnancy at ≥37 weeks of amenorrhea, instrumental delivery |
| | 4a forceps delivery |
| | 4b spatula delivery |
| | 4c vacuum delivery |
| 5 | All women with a single cephalic pregnancy at <37 weeks of amenorrhea |
| 6 | All women with a single breech pregnancy |
| 7 | All women with multiple pregnancy |

## Statistical analysis

Characteristics of women were described for the 3 ENP surveys (2010, 2016, 2021). First, for each edition of the ENP, we calculated the overall prevalence of episiotomy and OASI. We then calculated the prevalence of episiotomy and OASI for each specific clinical context of the previously defined seven-group classification (i.e., number of women with episiotomy or OASI out of the total number of women in the group).

Trends in both episiotomy and OASI prevalences for 2010, 2016, and 2021 were analyzed overall and by group (except in groups 4a to 7 where the number of OASIs was very low) using Cochran–Armitage tests, with the year 2010 as reference. In order to take into account confounding factors related to changes in population characteristics, we performed multivariable Poisson regression analyses with robust variance adjusted for maternal age, BMI, country of birth, antenatal classes, suspicion of fetal macrosomia, neuroaxial analgesia during labor, the professional who attended the birth, the annual number of deliveries, and the maternity unit status. These factors were selected on a priori hypotheses according to the literature, on author's clinical experiences and on the results of the descriptive analysis. The professional who attended the birth was not introduced into the model for the analysis of groups 2a, 2b, 2c, 4a, 4b, and 4c because instrumental delivery must be done by obstetricians in France. Crude and adjusted risk ratios (aRRs) were estimated with their confidence intervals (CI). Finally, we described the variations in episiotomy rates by the annual number of deliveries and the maternity unit status. We also described the relative size of each group (i.e., number of women in the group divided by total number of women delivered), and each group's contribution to both episiotomy and OASI prevalence (i.e., number of episiotomies or OASIs in the group divided by the total number of women having episiotomies or OASIs).

Data were missing for 8.3% of the studied women, justifying analyses using multiple imputation by chained equations with the STATA "mi impute" procedure. We generated 10 independent imputed data sets. Imputation model variables included the maternal and obstetric

characteristics introduced in the final model and outcomes. Estimates were pooled according to Rubin's rule [30]. Statistical significance was set at a two-tailed test with $P < 0.05$. All calculated prevalences, and the group's contribution and relative size, were presented with their 95% CI. Analyses were performed with Stata 16.0 software. This study is reported as per the Strengthening the Reporting of Observational Studies in Epidemiology (STROBE) guidelines (S1 STROBE Checklist) [31].

## Results

The study population included 29,750 women (Fig 1).

Between 2010 and 2021, women's characteristics changed: they were older at delivery, had a higher BMI, a higher level of education, and fetal macrosomia was more frequently suspected. Women also gave birth less frequently in private maternity hospitals. Finally, the proportion of instrumental deliveries remained stable from 14.5% in 2010 to 15.8% in 2021. However, the type of instrument used for delivery changed: the proportion of vacuum deliveries increased from 6.4% to 9.5%, while the proportion of forceps and spatula deliveries decreased from 4.7% to 3.3% and 3.4% to 3.0%, respectively (Table 1).

The overall prevalence of episiotomy significantly decreased from 25.8% (95% CI 25.0 to 26.7) in 2010 to 20.1% (95% CI 19.3 to 20.9) in 2016 and 8.3% in 2021 (95% CI 7.8 to 8.9) (Cochran–Armitage tests, $P < 0.001$). When taking into account the maternal and obstetric characteristics, the aRR of episiotomy decreased by 22% between 2010 and 2016 (aRR 0.78, 95% CI 0.75 to 0.82), and by 67% between 2010 and 2021 (aRR 0.33, 95% CI 0.30 to 0.35).

Analyses performed according to the seven-group classification showed a decrease in the rate of episiotomy in all groups (Cochran–Armitage tests, $P < 0.001$) (Table 2). The most significant decreases were observed in Group 3 [multiparous women with a singleton in cephalic

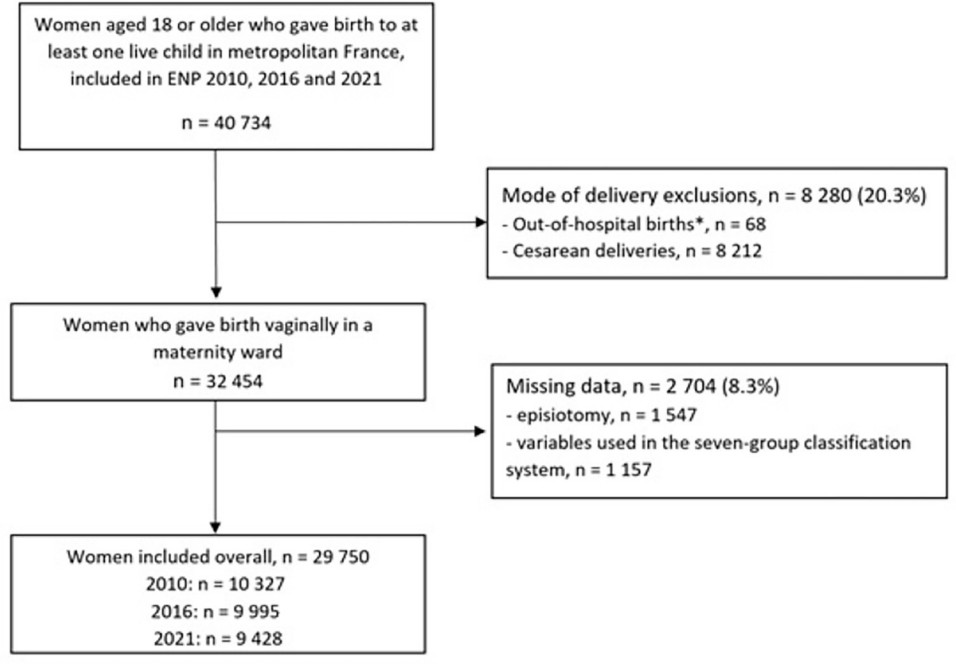

**Fig 1. Flow chart.**

**Table 1. Trends in the characteristics of women giving birth vaginally in France between 2010 and 2021.**

| Characteristics, *n* (%) | 2010 *N* = 10,327 | 2016 *N* = 9,995 | 2021 *N* = 9,428 |
|---|---|---|---|
| Maternal age, years | | | |
| <25 | 1,674 (16.2) | 1,385 (13.9) | 1,131 (12.0) |
| 25–35 | 6,680 (64.7) | 6,634 (66.4) | 6,198 (65.7) |
| >35 | 1,883 (18.2) | 1,968 (19.7) | 2,099 (22.3) |
| BMI before pregnancy, kg/m$^2$ | | | |
| <18.5 | 842 (8.1) | 737 (7.4) | 526 (5.6) |
| 18.5–24.9 | 6,413 (62.1) | 5,804 (58.1) | 5,003 (53.1) |
| 25–29.9 | 1,612 (15.6) | 1,770 (17.7) | 1,849 (19.6) |
| ≥30 | 844 (8.2) | 942 (9.4) | 1,070 (11.3) |
| Country of birth | | | |
| France | 8,222 (79.6) | 7,734 (77.4) | 6,835 (72.5) |
| Others | 1,756 (17.0) | 1,646 (16.5) | 1,667 (17.7) |
| Antenatal classes | | | |
| No | 5,345 (51.8) | 4,341 (43.4) | 3,841 (40.7) |
| Yes | 4,667 (45.2) | 4,997 (50.0) | 4,690 (49.8) |
| Parity | | | |
| Primiparous | 3,788 (36.7) | 4,102 (41.0) | 3,765 (39.9) |
| Multiparous | 6,539 (63.3) | 5,893 (59.0) | 5,663 (60.1) |
| Mode of delivery | | | |
| Vaginal delivery | 8,835 (85.5) | 8,451 (84.5) | 7,935 (84.2) |
| Forceps delivery | 482 (4.7) | 427 (4.3) | 312 (3.3) |
| Spatula delivery | 353 (3.4) | 351 (3.5) | 284 (3.0) |
| Vacuum delivery | 657 (6.4) | 766 (7.7) | 897 (9.5) |
| History of caesarean section | | | |
| No | 9,634 (93.3) | 9,404 (94.1) | 8,863 (94.0) |
| Yes | 532 (5.1) | 588 (5.9) | 562 (6.0) |
| Suspicion of fetal macrosomia | | | |
| No | 9,940 (96.2) | 9,554 (95.6) | 8,163 (86.6) |
| Yes | 299 (2.9) | 386 (3.9) | 672 (7.1) |
| Analgesia during labor | | | |
| No | 2,275 (22.0) | 1,785 (17.9) | 1,541 (16.3) |
| Epidural anesthesia | 8,019 (77.7) | 8,174 (81.8) | 7,820 (82.9) |
| Others | 29 (0.3) | 13 (0.1) | 61 (0.7) |
| Gestational age at birth, weeks | | | |
| <37 | 492 (4.8) | 530 (5.3) | 421 (4.5) |
| [37–41] | 7,895 (76.4) | 7,697 (77.0) | 7,251 (76.9) |
| ≥41 | 1,940 (18.8) | 1,768 (17.7) | 1,756 (18.6) |
| Birthweight, g | | | |
| <2,500 | 433 (4.2) | 497 (5.0) | 398 (4.2) |
| [2,500–3,500] | 6,362 (61.6) | 6,186 (61.9) | 5,644 (59.9) |
| [3,500–4,000] | 2,822 (27.3) | 2,659 (26.6) | 2,625 (27.9) |
| ≥4,000 | 700 (6.8) | 646 (6.4) | 645 (6.8) |
| Professional who attended the delivery | | | |
| Obstetrician | 3,006 (29.1) | 2,529 (25.3) | 2,255 (23.9) |
| Midwife | 6,864 (66.5) | 6,913 (69.2) | 6,320 (67.0) |
| Annual number of deliveries | | | |

*(Continued)*

**Table 1.** (Continued)

| Characteristics, *n* (%) | 2010 N = 10,327 | 2016 N = 9,995 | 2021 N = 9,428 |
|---|---|---|---|
| <1,500 | 4,013 (38.9) | 3,358 (33.6) | 3,292 (34.9) |
| [1,500–3,499] | 4,433 (42.9) | 3,732 (37.3) | 3,208 (34.0) |
| ≥3,500 | 1,881 (18.2) | 2,904 (29.1) | 2,928 (31.1) |
| Maternity unit status | | | |
| University hospitals | 1,784 (17.3) | 1,924 (19.2) | 1,900 (20.1) |
| Public hospitals | 5,698 (55.2) | 5,813 (58.2) | 5,540 (58.8) |
| Private hospitals | 2,845 (27.5) | 2,257 (22.6) | 1,988 (21.1) |

**Table 2. Variations in the prevalence of episiotomy in France between 2010 and 2021 according to the classification for episiotomy practices.**

| Groups | 2010 | | 2016 | | | 2021 | | | P-Trend* |
|---|---|---|---|---|---|---|---|---|---|
| | *n/N* | Episiotomy % (95% CI) | *n/N* | Episiotomy % (95% CI) | aRR$ [95% CI] | *n/N* | Episiotomy % (95% CI) | aRR$ [95% CI] | |
| 1-Nulliparous women, singleton, cephalic, at term, non-instrumental delivery | 878/2,479 | 35.4 (33.5–37.3) | 674/2,688 | 25.1 (23.4–26.8) | 0.73 [0.67–0.79] | 228/2,443 | 9.3 (8.2–10.6) | 0.27 [0.23–0.31] | <0.001 |
| 2a-Nulliparous women, singleton, cephalic, at term, forceps delivery** | 300/355 | 84.5 (80.3–87.9) | 254/313 | 81.2 (76.4–85.1) | 0.98 [0.90–1.05] | 139/248 | 56.0 (49.8–62.1) | 0.67 [0.59–0.77] | <0.001 |
| 2b-Nulliparous women, singleton, cephalic, at term, spatula delivery** | 190/241 | 78.8 (73.2–83.6) | 178/253 | 70.4 (64.4–75.7) | 0.88 [0.78–0.98] | 64/202 | 31.7 (25.6–38.5) | 0.39 [0.30–0.49] | <0.001 |
| 2c-Nulliparous women, singleton, cephalic, at term, vacuum delivery** | 252/449 | 56.1 (51.4–60.8) | 241/539 | 44.7 (40.5–49.0) | 0.79 [0.70–0.90] | 163/635 | 25.7 (22.3–29.3) | 0.45 [0.39–0.53] | <0.001 |
| 3-Multiparous women, singleton, cephalic, at term, non-instrumental delivery | 694/5,764 | 12.0 (11.2–12.9) | 399/5,161 | 7.7 (7.0–8.5) | 0.65 [0.58–0.73] | 117/5,036 | 2.3 (1.9–2.8) | 0.20 [0.16–0.24] | <0.001 |
| 4a-Multiparous women, singleton, cephalic, at term, forceps delivery** | 76/109 | 69.7 (60.3–77.7) | 50/90 | 55.6 (50.0–65.6) | 0.75 [0.59–0.95] | 13/47 | 27.7 (16.5–42.6) | 0.36 [0.22–0.61] | <0.001 |
| 4b-Multiparous women, singleton, cephalic, at term, spatula delivery** | 59/84 | 70.2 (59.4–79.2) | 37/72 | 51.4 (39.7–62.9) | 0.67 [0.50–0.90] | 11/63 | 17.5 (9.7–29.2) | 0.27 [0.14–0.50] | <0.001 |
| 4c-Multiparous women, singleton, cephalic, at term, vacuum delivery** | 52/179 | 29.1 (22.5–36.3) | 57/188 | 30.3 (23.8–37.4) | 0.97 [0.70–1.34] | 22/235 | 9.4 (6.0–13.8) | 0.30 [0.19–0.47] | <0.001 |
| 5- Singleton, cephalic, <37 WG | 79/422 | 18.7 (15.1–22.8) | 61/434 | 14.1 (10.9–17.7) | 0.72 [0.54–0.98] | 18/379 | 4.7 (2.8–7.4) | 0.26 [0.16–0.41] | <0.001 |
| 6- Singleton breech pregnancy | 39/73 | 53.4 (41.4–65.2) | 21/60 | 35.0 (23.1–48.4) | 0.51 [0.34–0.77] | 11/56 | 19.6 (10.2–32.4) | 0.23 [0.13–0.41] | <0.001 |
| 7- Multiple pregnancy | 45/172 | 26.2 (19.8–33.4) | 40/197 | 20.3 (14.9–26.6) | 0.94 [0.65–1.38] | 1/84 | 1.2 (0.0–6.5) | 0.06 [0.01–0.40] | <0.001 |
| Total | 2,664/10,327 | **25.8 (25.0–26.7)** | 2,012/9,995 | **20.1 (19.3–20.9)** | 0.78 [0.75–0.82] | 787/9,428 | **8.3 (7.8–8.9)** | 0.33 [0.30–0.35] | <0.001 |

The reference year is 2010. WG: weeks of gestation aRR: Adjusted risk ratios.

* Overall trend test across 2010 to 2021 using Cochran–Armitage test.

$ Adjusted risk ratios obtained after multiple imputation from Poisson regression models with robust variance estimation, adjusted for maternal age, BMI, country of birth, antenatal classes, suspicion of fetal macrosomia, neuro-axial analgesia during labor, professional who attended the delivery, and maternity unit size and status.

**For the analysis of groups 2a, 2b, 2c, 4a, 4b, and 4c, the variable "Professional who attended the delivery" was not introduced into the model because instrumental delivery can only be used by obstetricians in France.

position at ≥37 weeks of amenorrhea and a spontaneous delivery] (aRR 0.20, 95% CI 0.16 to 0.24) and in Group 7 [multiple pregnancy] (aRR 0.06, 95% CI 0.01 to 0.40). The lowest decrease in the episiotomy rate was in Group 2a [nulliparous women with a singleton in cephalic position at ≥37 weeks of amenorrhea with forceps delivery] (aRR 0.67, 95% CI 0.59 to 0.77). Crude RR are presented in supplemental data (S1 Table). Secondary analyses showed that the decrease in the overall prevalence of episiotomy was observed whatever the annual number of deliveries in the maternity unit and the maternity unit status (S2 and S3 Tables).

The overall prevalence of OASI increased from 0.7% (95% CI 0.6 to 0.9) in 2010 to 0.9% (95% CI 0.7 to 1.1) in 2016 and 1.0% (95% CI 0.8 to 1.3) in 2021 (Cochran–Armitage test, $P$ = 0.021), but this increase was statistically nonsignificant after adjustment for the characteristics of mothers and maternity units (aRR 1.24, 95% CI 0.91 to 1.68) (Table 3). Crude RR are presented in supplemental data (S4 Table). Analyses using the seven-group classification revealed a significant increase in the prevalence of OASI within Group 2b [Nulliparous women, singleton, cephalic, at term, spatula delivery], from 2.6% (95% CI 1.2 to 5.6) in 2010 to 9.6% (95% CI 6.2 to 14.7) in 2021 (aRR 3.69, 95% CI 1.50 to 9.09). The prevalence of OASI within Group 2a

**Table 3. Variations in the prevalence of OASI in France between 2010 and 2021 according to the classification for episiotomy practices.**

| Groups | 2010 | | 2016 | | | 2021 | | | P-Trend* |
|---|---|---|---|---|---|---|---|---|---|
| | n / N | OASI % (95% CI) | n / N | OASI % (95% CI) | aRR$ [95% CI] | n/N | OASI % (95% CI) | aRR$ [95% CI] | |
| 1-Nulliparous women, singleton, cephalic, at term, non-instrumental delivery | 23/2,449 | 0.9 (0.6–1.6) | 30/2,610 | 1.1 (0.8–1.6) | 1.14 [0.66–1.98] | 27/2,429 | 1.1 (0.7–1.7) | 1.10 [0.64–1.90] | 0.551 |
| 2a-Nulliparous women, singleton, cephalic, at term, forceps delivery** | 11/342 | 3.2 (1.8–5.7) | 12/290 | 4.1 (2.4–7.2) | 1.13 [0.49–2.60] | 14/244 | 5.7 (3.4–9.5) | 1.78 [0.81–3.90] | 0.144 |
| 2b-Nulliparous women, singleton, cephalic, at term, spatula delivery** | 6/233 | 2.6 (1.2–5.6) | 12/220 | 5.5 (3.1–9.4) | 2.16 [0.82–5.64] | 19/197 | 9.6 (6.2–14.7) | 3.69 [1.50–9.09] | 0.002 |
| 2c-Nulliparous women, singleton, cephalic, at term, vacuum delivery** | 10/446 | 2.2 (1.1–4.1) | 7/508 | 1.4 (0.6–2.8) | 0.59 [0.22–1.55] | 11/625 | 1.8 (0.9–3.1) | 0.74 [0.31–1.76] | 0.578 |
| 3-Multiparous women, singleton, cephalic, at term, non-instrumental delivery | 19/728 | 0.3 (0.2–0.5) | 10/5,119 | 0.2 (0.1–0.4) | 0.54 [0.24–1.19] | 16/5,020 | 0.3 (0.2–0.5) | 0.75 [0.37–1.53] | 0.803 |
| 4a-Multiparous women, singleton, cephalic, at term, forceps/spatula delivery | 2/190 | 1.1 (0.1–3.8) | 2/151 | 1.3 (0.2–4.7) | NC | 5/110 | 4.5 (2.5–10.3) | NC | NC |
| 4a-Multiparous women, singleton, cephalic, at term, forceps delivery | 1/107 | NC | 1/87 | NC | NC | 3/47 | NC | NC | NC |
| 4b-Multiparous women, singleton, cephalic, at term, spatula delivery* | 1/83 | NC | 1/64 | NC | NC | 2/63 | NC | NC | NC |
| 4c-Multiparous women, singleton, cephalic, at term, vacuum delivery** | 1/177 | 0.6 (0.0–3.1) | 2/182 | 1.1 (0.1–3.9) | NC | 4/231 | 1.7 (0.5–4.4) | NC | NC |
| 5-Singleton, cephalic, <37 WG | 0/420 | 0.0 (0.0–0.9) | 4/427 | 0.9 (0.3–2.4) | NC | 0/377 | 0.0 (0.0–1.0) | NC | NC |
| 6-Singleton breech pregnancy | 0/71 | 0.0 (0.0–5.1) | 1/55 | 1.8 (0.0–9.7) | NC | 0/56 | 0.0 (0.0–6.4) | NC | NC |
| 7-Multiple pregnancy | 2/170 | 1.1 (0.1–4.2) | 3/193 | 1.6 (0.3–4.5) | NC | 1/83 | 1.2 (0.0–6.5) | NC | NC |
| Total | 74/10,226 | 0.7 (0.6–0.9) | 83/9,755 | 0.9 (0.7–1.1) | 1.08 [0.79–1.49] | 97/9,372 | 1.0 (0.8–1.3) | 1.24 [0.91–1.68] | 0.021 |

The reference year is 2010 for all analyses. WG: weeks of gestation. NC: Not calculated. OASI: obstetric anal sphincter injury. aRR: Adjusted risk ratios.

*Overall trend test across 2010 to 2021 using Cochran–Armitage test.

$Adjusted risk ratios obtained after multiple imputation from Poisson regression models with robust variance estimation, adjusted for maternal age, BMI, country of birth, antenatal classes, suspicion of fetal macrosomia, neuro-axial analgesia during labor, professional who attended the delivery, and maternity unit size and status.

**For the analysis of groups 2a, 2b, 2c, 4a, 4b, and 4c, the variable "Professional who attended the delivery" was not introduced into the model because instrumental delivery can only be used by obstetricians in France.

[Nulliparous women, singleton, cephalic, at term, forceps delivery] increased from 3.2% (95% CI 1.8 to 5.7) in 2010 to 5.7% (95% CI 3.4 to 9.5) in 2021, but this increase was statistically non-significant after adjustment (aRR 1.78, 95% CI 0.81 to 3.90) (Table 3). These 2 groups accounted for more than one third of OASIs in 2021 (S5 Table).

## Discussion

From 2010 to 2021, the overall prevalence of episiotomy in France significantly decreased three-fold, while the overall prevalence of OASI did not significantly increase. The proportions of spontaneous vaginal deliveries and instrumental extractions remained stable, but there was a change in the type of instruments used, with an increasing use of vacuum and a decreasing use of forceps and spatula. Analyses by obstetrical clinical contexts, using the seven-group classification, revealed specific results for nulliparous women who gave birth to a full-term infant in a cephalic position with spatula (Group 2b). In this group, the prevalence of episiotomy was reduced by more than half (from 78.8% to 31.7%) while the prevalence of OASI tripled (from 2.6% to 9.6%).

Despite international guidelines encouraging the limited use of episiotomy, the overall prevalence of episiotomy varies considerably between and within countries. In some countries, the overall prevalence is still currently high [32,33], or even very high [34], and implementing evidence-based obstetric practices remains a major challenge [35,36]. In France, the overall prevalence of episiotomy is less than 10%, which is in line with WHO recommendations [37] and comparable to the rate observed in other countries where a very restrictive episiotomy policy is applied [38]. The accelerated reduction towards the end of the 2020s in France could be explained by several factors: successive national guidelines promoting a restrictive policy on episiotomies in all childbirth contexts, even for instrumental deliveries [12,22,23,39], audits on the use of episiotomies carried out in maternity units and by regional perinatal care networks [20], and finally more women requesting not to have an episiotomy [40].

We found that the steep overall drop in episiotomy rates in France was not followed by a marked increase in OASI. The overall prevalence of OASI increased slightly to around 1% in 2021, but the increase was statistically nonsignificant and the prevalence remains very low compared with many other countries [38]. Manual perineal protection, which is known to reduce the risk of OASI [41] and is almost systematically performed in France [42], may partly explain the low prevalence of OASI. A review of the literature has shown conflicting conclusions [43]. Some studies reported a stable prevalence of OASI over time, while others observed moderate to substantial increases concurrent to decreasing episiotomy rates [17,18,44–46].

Our findings are reassuring considering that we did not detect significant increases in the overall prevalence of OASI, but trends in the prevalences of episiotomy and OASI should be interpreted by subgroups of women to better account for the diversity of obstetric contexts. For instance, among nulliparous women with single cephalic delivery at term with forceps (Group 2a) and spatula (Group 2b), the prevalence of OASI doubled or even tripled in case of spatula delivery. Two likely interconnected factors may have contributed to this increase. First, we observed a decrease in the use of forceps and spatula in favor of vacuum delivery during the studied period, which is in line with French guidelines recommending the use of vacuum extraction as a first-line protection against OASI [16]. It is therefore possible that forceps and spatula were increasingly used in more complex obstetric situations, leading to an increased risk of OASI in 2021 compared to 2016. Secondly, there was a steep reduction in episiotomy prevalence in these 2 groups, and the rates of episiotomy became low in 2021. We found a statistically significant association only in spatula deliveries, where the reduction in the prevalence of episiotomy was more marked than in forceps deliveries. Nonetheless, given the small number of OASI in forceps deliveries, the results should be considered with caution. Contrary

to other international guidelines [14,15], the 2018 French guidelines recommended a restricted use of episiotomy during vaginal delivery, including instrumental delivery [16]. Considering the increased prevalence of OASI observed in our study, and in light of accumulating new evidence from observational studies published after 2018 [41,44–47] suggesting a protective role of episiotomy against OASI in instrumental deliveries, it may be worth reconsidering the substantial reduction in episiotomy rates in populations at a high risk of OASI and revising the indications for episiotomy in specific contexts involving instrumental deliveries in the French guidelines.

Another important finding is the stable prevalence (2%) of OASI in nulliparous women who delivered a full-term infant in the cephalic position with vacuum (Group 2c), even though episiotomy use was halved in this group. The more frequent use of vacuum-assisted delivery may also have influenced the overall prevalence of OASI since vacuum-assisted delivery is associated with a reduced risk of OASI compared with forceps or spatula delivery [37]. However, these results should be interpreted with caution due to the low prevalence of OASI in our study compared to the literature [48]. Additionally, accumulating evidence from both observational studies [44,46,47] and a recent multicenter, open label, randomized controlled trial [49] found a significant reduction in OASI with lateral episiotomy compared to no episiotomy. Finally, the prevalence of episiotomy in nulliparous women who spontaneously gave birth to a full-term infant in the cephalic position (Group 1) decreased from 25% to 9% without significantly increasing the prevalence of OASI. This result leads us to believe that women at risk of OASI were probably correctly identified. Such results should encourage midwives and obstetricians to reduce the use of episiotomy in hospitals or countries where the prevalence is high. However, the optimal rate of episiotomy remains unknown, and episiotomy can still be useful for indications such as fetal distress [50]. Finally, we cannot exclude the hypothesis that further reductions in the use of episiotomy could lead to an increase in OASI, and similar analyses should therefore be conducted in the future.

This study has several strengths. The last 3 ENP surveys followed the same design and methodology. The variations in the prevalences of both episiotomy and OASI were studied concomitantly and according to a clinically relevant classification of obstetric contexts specifically built to assess variations in the risk of episiotomy and OASI, providing new knowledge and opening up new perspectives for clinical practice and research. The source population included large, representative national samples of births in France, making it possible to study rare outcomes such as OASI and to take risk factors into account. The database provided detailed information on the sociodemographic, obstetric, and organizational characteristics relevant for the study of changes in episiotomy use and OASI. For each survey, specially trained research midwives interviewed women during the postpartum stay and collected information from their medical records. An audit system was implemented to guarantee both the completeness and the quality of the data collected.

Nonetheless, this study presents some limitations. The data collected for OASI did not distinguish between third- and fourth-degree tears, which do not have the same long-term consequences. Information about failed attempts of instrumental delivery was not collected although it would have been useful to investigate as a means of potentially explaining higher rates of OASI in some groups. Spatula deliveries account for about 20% of instrumental births in France. This technique is also used elsewhere in Europe and in Latin America [51], but remains rare in Anglo-Saxon countries, limiting the generalizability of this finding.

Despite the large size of our population, we were not able to analyze OASI trends in all groups given its low frequency. Indeed, analyses on OASI were performed in the 4 main groups of women, corresponding to 91.5% of the whole study population (26,851/29,353). Although we used a published classification system representing obstetric contexts with

significant risk factors for perineal issues, potential factors such as suspected fetal macrosomia were not considered, but they were included as adjustment factors in our models wherever possible. The mode of onset of labor was not included in our models to avoid an overadjustment of risk with macrosomia and because the rate remained stable during the study period (22.7% in 2010, 22.0% in 2016, and 25.8% in 2021) [27]. It would also have been useful to consider other factors, such as duration of expulsive efforts, fetal head position, and station at the beginning of active second stage, but these data were not collected in all 3 editions of the ENP.

A restrictive episiotomy policy has been successfully implemented in France for both spontaneous and instrumental deliveries, and was not followed by an overall increase in OASI. However, subgroup analyses revealed a significant increase in OASI in nulliparous women giving birth by spatula (Group 2b). In this group, the prevalence of episiotomy was reduced by more than half, while the prevalence of OASI tripled. A clinically relevant but nonsignificant increase was observed in the prevalence of OASI in nulliparous women delivering with forceps (Group 2a). No increase in OASI was observed in nulliparous women delivering with vacuum (Group 2c). Results in subgroups 2a and 2c should be interpreted with caution given the low prevalence of OASI and potential lack of statistical power. Further studies with a high level of evidence are needed to predict the optimal rate of episiotomy for this group of women at high risk of OASI. Finally, our findings suggest that episiotomy use can be safely reduced for spontaneous vaginal deliveries, particularly in hospitals and regions where episiotomy rates remain high to comply with international guidelines and women's requests.

## Supporting information

**S1 STROBE Checklist. STROBE Statement—Checklist of items that should be included in reports of cross-sectional studies.**
(DOCX)

**S1 Table. Variations in the prevalence of episiotomy in France between 2010 and 2021 according to the classification for episiotomy practices (Crude Risk Ratio).**
(DOCX)

**S2 Table. Variations in the prevalence of episiotomy in France between 2010 and 2021 according to the annual number of deliveries.**
(DOCX)

**S3 Table. Variations in the prevalence of episiotomy in France between 2010 and 2021 according to the status of maternity unit.**
(DOCX)

**S4 Table. Variations in the prevalence of OASI in France between 2010 and 2021 according to the classification for episiotomy practices (Crude Risk Ratio).**
(DOCX)

**S5 Table. Variations in the relative size of each group, the contribution of episiotomy and OASI prevalence according to the classification for episiotomy practices.**
(DOCX)

## Acknowledgments

We thank the Maternal and Child Health Services in each district, regional perinatal networks, and regional health agencies without which these surveys could not have been conducted. We

also acknowledge all the local investigators who rigorously collected the data in each maternity unit, as well as all the mothers who agreed to be interviewed. We thank the supervisors of all the maternity units (department heads and midwife coordinators) who agreed to have the survey performed in their department. The authors would like to thank Suzanne Rankin for editing and proofreading the manuscript.

## Author Contributions

**Conceptualization:** Anne Alice Chantry, Thomas Desplanches.

**Formal analysis:** Raphaele Houlbracq.

**Investigation:** Camille Le Ray, Béatrice Blondel, Nathalie Lelong.

**Project administration:** Camille Le Ray, Béatrice Blondel, Nathalie Lelong.

**Supervision:** Anne Alice Chantry, Thomas Desplanches.

**Writing – original draft:** Raphaele Houlbracq, Anne Alice Chantry, Thomas Desplanches.

**Writing – review & editing:** Raphaele Houlbracq, Camille Le Ray, Béatrice Blondel, Nathalie Lelong, Thomas Desplanches.

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
