## [Editor Report · Decision Letter 0]

26 Apr 2024

Dear Dr DESPLANCHES, 

Thank you for submitting your manuscript entitled "IMPLEMENTATION OF A RESTRICTIVE EPISIOTOMY POLICY: RESULTS FROM THE FRENCH NATIONAL PERINATAL SURVEYS IN 2010, 2016 AND 2021" for consideration by PLOS Medicine.

Your manuscript has now been evaluated by the PLOS Medicine editorial staff as well as by an academic editor with relevant expertise and I am writing to let you know that we would like to send your submission out for external peer review.

Please re-submit your manuscript within two working days, i.e. by Apr 30 2024 11:59PM.

Kind regards,

Katrien G. Janin, PhD

Senior Editor

PLOS Medicine

---

## [Decision Letter · Decision Letter 1]

21 Jun 2024

Dear Dr DESPLANCHES,

Many thanks for submitting your manuscript "IMPLEMENTATION OF A RESTRICTIVE EPISIOTOMY POLICY: RESULTS FROM THE FRENCH NATIONAL PERINATAL SURVEYS IN 2010, 2016 AND 2021" (PMEDICINE-D-24-01340R1) to PLOS Medicine. The paper has been reviewed by subject experts and a statistician; their comments are included below and can also be accessed here: [LINK]

After discussing the paper with the editorial team and an academic editor with relevant expertise, I'm pleased to invite you to revise the paper in response to the reviewers' comments. We plan to send the revised paper to some or all of the original reviewers, and we cannot provide any guarantees at this stage regarding publication.

We ask that you submit your revision by Jul 12 2024 11:59PM. However, if this deadline is not feasible, please contact me by email, and we can discuss a suitable alternative.

Don't hesitate to contact me directly with any questions (kjanin@plos.org). 

Best regards, 

Katrien 

Katrien Janin, PhD 

Associate Editor

PLOS Medicine

kjanin@plos.org

Comments from the academic editor:

I suggest that to accommodate a broader audience an explanation of restrictive episiotomy and the context of its introduction of France would be beneficial. In order to make these results generalisable to the world, a more detailed discussion about practices in France would be advisable. 

Comments from the reviewers: 

Reviewer #1: Implementation of a Restrictive Episiotomy Policy: Results from the French National Perinatal Surveys in 2010, 2016 and 2021

The authors addressed a challenging issue in the field of women's health - decreasing the prevalence of episiotomy with no increase in prevalence of obstetric anal sphincter injury (OASI). The authors assessed the prevalence of both episiotomy and OASI, after the implementation of a restrictive episiotomy policy. Results show significant reduction in prevalence of episiotomy with no signigificant increase in OASI. Findings can give insights in this challenging issue, giving room to selective episiotomy for women with specific charateristics and also to continuous quality improvement in this field.

- A nationwide sample covering three time periods and standardized procedures for data collection were used.

- Authors sought ethical approval.

- Abstract conveys the main results of the study.

- Statistical approach is apprpriate

- Authors cited recent appropriate research and guidelines

Few points deserve attention and should be explained in more detail. The following are the comments, questions and suggestions: 

Abstract

The authors stated: "The significant reduction in the prevalence of episiotomy in France has not been followed by an increase in OASI, except in nulliparous women giving birth by forceps/spatula (Group 2a)." Due to the small number of events, there are results for only 4 groups/subgroups (out of all 9 groups /subgroups). This should appear in a clear way in the conclusion section.

Methods

1 - For readers out of France, it would be welcome an explanation in brief about how French healthcare system (maternity units) works. For example: are there private healthcare services? 

2 - What is the refusal rate for each edition of the ENP? 

3 - Authors should explain the procedures used for statistical modelling. What are the rules to keep variables in the regression models? All independent variables were forced in the models?

Results

1- In the Table 1: meaning of P-value* doesn´t appear at the bottom of the table.

2 - Authors wrote: "The highest decrease was observed in Group 3 [multiparous women with a singleton in cephalic position at ≥37 weeks of amenorrhea and a spontaneous delivery] (aRR 0.20, 95% CI 0.16-0.24)". According to the Table 2, this is true after excluding group 7.

3 - In regards to the S1 Table and S2 Table, I don't understand the information at the bottom: "Adjusted risk ratios obtained after multiple imputation from Poisson regression models with robust variance estimation, adjusted for maternal age, BMI, country of birth, antenatal classes, suspicion of fetal macrosomia, neuro-axial analgesia during labor, professional who attended the delivery**, and maternity unit size and status." I suppose the number of deliveries is an indicator of maternity unit size. Why did authors adjust for maternity unit size if analysis was stratified by this variable? The same for the variable maternity unit status. Moreover, if these Tables display results for all groups as a whole, I don't understand why authors wrote at the bottom: "**Professional who attended the delivery was not introduced into the model for the analysis of groups 2a, 2b, 4a and 4b because instrumental delivery can only be used by obstetricians in France."

Discussion/Conclusion

Authors should be cautious in regards to their conclusions about variation in prevalence of OASI. They stated: "Analyses using a seven-group classification system showed that the drop in episiotomy use across all groups not followed by a notable increase in OASI, except among nulliparous women delivering with forceps/spatula." However, it is important to say they were not able to analyze OASI trends in all groups due to the small number of events. Indeed, there are results for only 4 groups/subgroups (out of all 9 groups /subgroups). This should appear in a clear way in the conclusion section.

Reviewer #2: This is a cross-sectional study of vaginal births >21 weeks' gestation in France at three time points 2010, 2016, 2021 with data obtained from a national perinatal survey. They aimed to assess temporal changes in episiotomy and obstetric anal sphincter injury (OASI) across these time points They split the population into seven clinically informative groups, assessed the distribution of relevant characteristics, the rate of episiotomy, and the rate of OASI, in each group across all three time points. They also used multivariable regression to assess adjusted relative changes in episiotomy and OASI rates in 2016 and 2021 vs 2010. Episiotomy decreased in all groups and OASI either decreased or was stable in all groups except for the nulliparous groups with spatula/forceps delivery. In the spatula forceps group, the episiotomy rate decreased by 50% and the OASI rate doubled. 

This is a well-written study with some interesting findings. There are some important issues that should be addressed.

1. The type of episiotomy (median, mediolateral, lateral) is not mentioned or discussed. This is a large omission given the well-established differences in the relationship between type of episiotomy and OASI risk. The type of episiotomy used, the angle of episiotomy (with mediolateral), and the potential for changes in these factors over time are critical to understanding these trends and their relationship with OASI trends. If these data are not available, provide more detail about episiotomy protocols in France (are they uniform?). 

2. It is useful to have separated the population into the seven clinically meaningful groups. These are very similar to the Robson 10-group classification for cesarean delivery and seem to be a modification of this system. It may be worthwhile to frame this classification as a modified Robson classification as it is very similar and the Robson classification has international recognition and uptake. 

3. While it is helpful to see these distinct groups it is not clear why births with spatula and forceps are presented together? These are two distinct instruments. Data from France regarding forceps are often not comparable to other contexts because they are aggregated with spatula in this way. Since most other countries do not use spatula, it is difficult to interpret/generalise. Unless one of these instruments are used far more than the other, there should be sufficient numbers to provide stable rates (out of a total of 835, 778, and 596 deliveries in 2010, 2016, 2021, resp.). If one group is far larger than the other (spatula often accounts for >85% of these combined spatula/forceps groups in other publication from France), than this should be clarified. The rates of OASI in these disaggregated groups should also be reported. It is possible the rates and trends of episiotomy and OASI are different in these two groups and it would be important to understand if the aggregated rates are reflective of the experience with either or both instruments. This disaggregation should appear in Table 1 and in all in the other results tables with group 2 being split into 2a - forceps, b-spatula, and c-vacuum and 4a-forceps, 4b-spatula, 4c-vacuum. As the forceps/spatula group have the highest rates of epis and OASI and the inverse association with the temporal trends (decreasing epis, increasing OASI) it is important to focus and be detailed in this group. 

4. Why was onset of labour (spontaneous/induced) not collected? The frequency of induction has certainly changed over this time period in many settings and may be related to the trends observed in this analysis. 

5. Do the restrictive episiotomy guidelines in France extend beyond spontaneous vaginal delviery to include instrumental vaginal delivery? This is not consistent with other national guidelines (RCOG, ANZOG), which specify that episiotomy should be considered with instrumental birth, especially with forceps delivery and among nullipara. If so, the appropriateness of the French guideline (and these inconsistencies with the aforementioned guidelines) should be discussed here. Especially in light of accumulating evidence from observational (Okeahialam NA, et al. Mediolateral/lateral episiotomy with operative vaginal delivery and the risk reduction of obstetric anal sphincter injury (OASI): A systematic review and meta-analysis. Int Urogynecol J. 2022;33(6):1393-405) and experimental (Bergendahl S. Vacuum extraction and pelvic floor injury: A randomized controlled trial and cohort studies. Stockholm, Sweden: Karolinska Institutet; 2024) studies showing >50% decreases in OASI with vs without mediolateral/lateral episiotomy. The fact that the rate is decreasing with forceps/spatula/and vacuum should give us pause. What is the evidence driving this decrease in episiotomy among instrumental deliveries?

6. It seems failed attempts of instrumental delivery have been excluded and not accounted for. This should be acknowledged and how this omission might change affect the study's results should be discussed. 

7. The <2% rate of OASI in vacuum delivery among nullipara is very low compared to other high-income settings and it is difficult to decipher the reasons for these relatively low rates of OASI with vacuum without further clinical details (length of labour, fetal station, fetal position). As a result, the finding of this analysis showing that decreased episiotomy were seen while rates of OASI were stable, should be interpreted with caution as the generalisability of these results are questionable. This should be further emphasised given the mounting evidence showing lower rates of OASI with episiotomy with vacuum birth, particularly in nullipara. 

One minor comment, consider changing the titles of the tables to remove the word "evolution". These estimates do not present an "evolution" of factors. This is an analysis of three fairly disparate snapshots in time and it is not clear that these trends are linear between these time points. 

Reviewer #3: Thanks for the opportunity to read your manuscript. My role is statistical reviewer, so I have focused on the design, data, and analysis that are presented. I have put general comments first, followed by questions relevant to a specific section of the manuscript (with a page/paragraph reference). 

This study presents changes in rates of episiotomies in women from France, over 2010, 2016, and 2021. Data is from a population-based based survey of all live births with live vaginal delivery in women 18 years or over. Third and fourth degree tears were considered as a secondary variable, and a range of covariates was collected from the participants, the birth, and the site where the birth took place. Changes in rates were considered stratified by a previously established 7-group classification that incorporates parity and delivery mode. Changes in the outcomes were analysed with a C-A test, and a Poisson regression model that included potential confounders of rates of the outcomes. I thought this was a clearly written paper with interesting findings, and apart from a few queries the analysis was appropriate. 

Given the size of the study, any comparison of characteristics between years is likely to yield a small p-value for even a minor difference. I would remove the p-values from the results. 

Was there any data available on how many women refused the survey each year?

An optional suggestion - the table are fairly large, and you might be able to use some visualisations to describe the data in a more effective way. 

P7, Paragraph 3. Should be 'multivariable', not 'multivariate'

P8, Paragraph 1. Should this be 'variation' instead of 'evolution'?

P8, Paragraph 2. What criteria was used to decide on 10 imputations?

---

* Please upload any figures associated with your paper as individual TIF or EPS files with 300dpi resolution at resubmission; please read our figure guidelines for more information on our requirements: http://journals.plos.org/plosmedicine/s/figures. While revising your submission, please upload your figure files to the PACE digital diagnostic tool, https://pacev2.apexcovantage.com/. PACE helps ensure that figures meet PLOS requirements. To use PACE, you must first register as a user. Then, login and navigate to the UPLOAD tab, where you will find detailed instructions on how to use the tool. If you encounter any issues or have any questions when using PACE, please email us at PLOSMedicine@plos.org.

* Please provide additional details regarding participant consent. In the ethics statement in the Methods and online submission information, please ensure that you have specified what type you obtained (for instance, written or verbal, and if verbal, how it was documented and witnessed). If your study included minors, state whether you obtained consent from parents or guardians. If the need for consent was waived by the ethics committee, please include this information."

FIGURES AND TABLES

SUPPLEMENTARY MATERIAL

REFERENCES

STUDY TYPE-SPECIFIC REQUESTS

OBSERVATIONAL STUDIES

* Abstract: Please include the study design, population and setting, number of participants, years during which the study took place (enrollment and follow up), length of follow up, and main outcome measures.

* Please ensure that the study is reported according to the STROBE (or appropriate STOBE extension) guideline (available from: https://www.equator-network.org/reporting-guidelines/strobe) and include the completed STROBE (or STROBE extension) checklist as Supporting Information. Please add the following statement, or similar, to the Methods: "This study is reported as per the Strengthening the Reporting of Observational Studies in Epidemiology (STROBE) guideline (S1 Checklist)." When completing the checklist, please use section and paragraph numbers, rather than page numbers. 

* or FOR POPULATION HEALTH/REGISTRY STUDIES: Please ensure that the study is reported according to the RECORD guideline (available from https://www.record-statement.org) and include the completed checklist as Supporting Information. Please add the following statement, or similar, to the Methods: "This study is reported as per the Reporting of Studies Conducted using Observational Routinely-Collected Data (RECORD) guideline (S1 Checklist)." When completing the checklist, please use section and paragraph numbers, rather than page numbers.

* For all observational studies, in the manuscript text, please indicate: (1) the specific hypotheses you intended to test, (2) the analytical methods by which you planned to test them, (3) the analyses you actually performed, and (4) when reported analyses differ from those that were planned, transparent explanations for differences that affect the reliability of the study's results. If a reported analysis was performed based on an interesting but unanticipated pattern in the data, please be clear that the analysis was data driven. 

* Please state in the Methods section whether the study had a prospective protocol or analysis plan. If a prospective analysis plan (from your funding proposal, IRB or other ethics committee submission, study protocol, or other planning document written before analyzing the data) was used in designing the study, please include the relevant document(s) with your revised manuscript as a Supporting Information file to be published alongside your study and cite it in the Methods section. A legend for this file should be included at the end of your manuscript. If no such document exists, please make sure that the Methods section transparently describes when analyses were planned, and when/why any data-driven changes to analyses took place. Changes in the analysis, including those made in response to peer review comments, should be identified as such in the Methods section of the paper, with rationale.

---

## [Decision Letter · Decision Letter 2]

27 Aug 2024

Dear Dr DESPLANCHES,

Many thanks for submitting your manuscript "IMPLEMENTATION OF A RESTRICTIVE EPISIOTOMY POLICY: RESULTS FROM THE FRENCH NATIONAL PERINATAL SURVEYS IN 2010, 2016 AND 2021" (PMEDICINE-D-24-01340R2) to PLOS Medicine. The paper has been re-reviewed by subject experts and a statistician; their comments are included below and can also be accessed here: [LINK]

After discussing the paper with the editorial team and an academic editor with relevant expertise, I'm pleased to invite you to revise the paper in response to the reviewers' comments. We plan to send the revised paper to some or all of the original reviewers, and we cannot provide any guarantees at this stage regarding publication.

We ask that you submit your revision by Sep 17 2024 11:59PM. However, if this deadline is not feasible, please contact me by email, and we can discuss a suitable alternative.

Don't hesitate to contact me directly with any questions (kjanin@plos.org). 

Best regards, 

Katrien 

Katrien Janin, PhD 

Associate Editor

PLOS Medicine

kjanin@plos.org

Comments from the reviewers: 

Reviewer #1: Title: IMPLEMENTATION OF A RESTRICTIVE EPISIOTOMY POLICY: RESULTS FROM THE FRENCH

NATIONAL PERINATAL SURVEYS IN 2010, 2016 AND 2021

I read the revised manuscript. The authors addressed all my comments and questions.

I have no more comments. In my opinion the manuscript should be published

Reviewer #2: The authors have done a nice job addressing the Reviewers' comments. The manuscript is improved. Still, in the interest of patient safety, the authors need to be clearer about the distinction between episiotomy practice in instrumental vaginal delivery and non-instrumental vaginal delivery and their results in these groups. The mention of these issues in the Discussion is appreciated, but this needs to be better articulated in the Abstract and in the Conclusion.

Abstract

It is stated that the rate of OASI in nullipara with forceps (Group 2a) increased but that this was not statistically significant. It should be noted that this is most likely due to a type II error. Given the 342 forceps deliveries in 2010 and 244 forceps deliveries in 2021, with an alpha=0.05 and a beta=0.80 (80% power) the smallest detectable difference would be an rate ratio (RR) of 2.90 (even before including confounders in the model). The observed increase in OASI from 3.2% to 5.7% represents a 55% relative increase OASI. While this was not statistically significant due to lack of power, this should be highlighted as a clinically significant increase. This may seem trivial in this unique population with VERY low rates of OASI with forceps delivery, in most places that use forceps delivery often, the rates of OASI among these deliveries are from 8-25%. A 55% increase in these rates would be enormously significant.

As a result, the forceps delivery group should not be included in this interpretation. For example, the conclusion of the Abstract states: "Analyses performed in the four main groups of women confirmed the main result, except for nulliparous women giving birth by spatula (Group 2b), for whom the rate of OASI significantly increased." Group 2a also did not confirm the main result - the study simply did not have enough power to support or refute this in group 2a. Group 2c (vacuum deliveries with nullipara) similarly suffers from power issues (a minimum detectable difference RR=2.65). Most readers will get their main takeaways from the Abstract and it is important that these nuances are not misrepresented here. The final sentence in the abstract should be specific to spontaneous vaginal deliveries. 

Introduction

In lines 7-9 of the introduction, there is no distinction between instrumental and non-instrumental vaginal delivery. International guidelines are consistent in recommending that episiotomy is restrictive with non-instrumental vaginal delivery but not with instrumental vaginal delivery. The UK and Australia have episiotomy rates ~90% with forceps delivery. This distinction is important. Obstetricians who are proficient in forceps delivery in most countries will almost always perform a mediolateral or lateral episiotomy with forceps delivery in nulliparous patients to avoid OASI. Suggesting that episiotomy rates should be decreased in this population is not supported by this study, is contradicted by other studies, and could result in an increase in these severe injuries. 

Conclusion 

Again, the first sentence again does not describe the differences in the non-instrumental and instrumental vaginal delivery groups and states that a drop in episiotomy "across all groups" was not followed by an overall increase in OASI. The second sentence also ignores the likely type II error and generalizes the non-instrumental vaginal delivery findings to all groups. There should be an explicit distinction between the confidence in the estimates in the instrumental and non-instrumental vaginal delivery groups.

Minor points

Note the test for linear trend is Cochran-Armitage not Cochrane-Armitage. 

Pg 5, Line 6 - the references 1-4 do not support the associations stated, for example, there is no association between episiotomy and PPH shown in these four references. Suggest reviewing these refs and citing the original research papers to support your opening statement.

Pg 8, Line 12 needs a reference.

Reviewer #3: Thanks for the revised manuscript and responses to my original review. The revised manuscript covers my original set of queries.

---

## [Decision Letter · Decision Letter 3]

11 Nov 2024

Dear Dr. DESPLANCHES,

Thank you very much for re-submitting your manuscript "IMPLEMENTATION OF A RESTRICTIVE EPISIOTOMY POLICY: RESULTS FROM THE FRENCH NATIONAL PERINATAL SURVEYS IN 2010, 2016 AND 2021" (PMEDICINE-D-24-01340R3) for review by PLOS Medicine.

I have discussed the paper with my colleagues and the academic editor and it was also seen again by one reviewer. I am pleased to say that provided the remaining editorial and production issues are dealt with we are planning to accept the paper for publication in the journal.

[LINK]

If you have any questions in the meantime, please contact me (lgaynor@plos.org) or the journal staff (plosmedicine@plos.org).  

We look forward to receiving the revised manuscript by Nov 18 2024 11:59PM.   

Sincerely,

Louise Gaynor-Brook, MBBS PhD

Senior Editor 

PLOS Medicine

plosmedicine.org

Requests from Editors:

Thank you for your patience with a longer assessment process than we anticipated, and apologies for the delay in providing you with an editorial decision. The list below appears rather lengthy, but some of these points are more minor points which should not require a substantial amount of time to attend to. 

General comments:

Please add 1-2 sentences about spatula deliveries with regard to French obstetric practice, and comment on the generalisability to other countries/settings (this can be added to the Discussion). 

Please note that line numbers referred to in this letter correspond to those in the attached manuscript file. 

To help us extend the reach of your research, please provide any Twitter handle(s) that would be appropriate to tag, including your own, your coauthors’, your institution, funder, etc. 

Title: Please revise your title according to PLOS Medicine's style. Please write in sentence case, rather than capitals throughout. Your title must be nondeclarative and not a question. It should begin with main concept if possible. Please place the study design in the subtitle (ie, after a colon). We suggest “Episiotomies and obstetric anal sphincter injuries following a restrictive episiotomy policy in France: An analysis of the 2010, 2016 and 2021 National Perinatal Surveys” or similar

Abstract Methods and Findings:

Please note that there is some flexibility in the Abstract word limit, which can be extended to allow the following edits to be incorporated. 

Please ensure that all numbers presented in the abstract are present and identical to numbers presented in the main manuscript text.

Please provide brief demographic details of the study population (e.g.age, ethnicity, important characteristics for this cohort, etc)

Please be consistent in the use of one decimal place for percentages. 

Line 51 - given that the difference in prevalence of OASI was not statistically significant, it would be better to state the prevalence in 2010 and 2021. We suggest “The difference in overall prevalence of OASI between 2010 (0.7%) and 2021 (1.0%) was not statistically significant after adjustment for… (aRR 1.24, 95% CI 0.91-1.68).” or similar. 

Please include the actual amounts and/or absolute risk(s) of relevant outcomes, not just aRRs.

Please include the important dependent variables that are adjusted for in the analyses.

Line 53 - please state the years being compared. 

Line 55 - please revise to ‘... and did not differ statistically significantly in Group 2a’ or similar.

Abstract Conclusions:

Line 63 - please revise to ‘statistically non-significant’

Line 67 - suggest revising to ‘...high episiotomy rates, our findings suggest that episiotomy could be safely…’

Author Summary:

In the final bullet point of ‘What Do These Findings Mean?’, please describe the main limitations of the study in non-technical language.

Methods:

Please ensure to include a separate subheading for your Methods section.

Did your study have a prospective protocol or analysis plan? Please state this (either way) early in the Methods section. If a prospective analysis plan (from your funding proposal, IRB or other ethics committee submission, study protocol, or other planning document written before analyzing the data) was used in designing the study, please include the relevant prospectively written document with your revised manuscript as a Supporting Information file to be published alongside your study, and cite it in the Methods section. A legend for this file should be included at the end of your manuscript. If no such document exists, please make sure that the Methods section transparently describes when analyses were planned, and if/when reported analyses differed from those that were planned. Changes in the analysis-- including those made in response to peer review comments-- should be identified as such in the Methods section of the paper, with rationale. If a reported analysis was performed based on an interesting but unanticipated pattern in the data, please be clear that the analysis was data-driven.

Ethics statement: Please provide details of ethical approval for your study (not just the ENPs). Please provide the name(s) of the institutional review board(s) that provided ethical approval. Please specify whether informed consent was written or oral.

Please ensure that the study is reported according to the STROBE guideline, and include the completed STROBE checklist as Supporting Information. Please add the following statement, or similar, to the Methods: "This study is reported as per the Strengthening the Reporting of Observational Studies in Epidemiology (STROBE) guideline (S1 Checklist)." The STROBE guideline can be found here: http://www.equator-network.org/reporting-guidelines/strobe/ When completing the checklist, please use section and paragraph numbers, rather than page numbers which will likely no longer correspond to the appropriate sections after copy-editing.

Results: 

Line 210 - unless statistical analyses have been performed, please revise “significantly changed” in reference to changes in women’s characteristics.

Line 263 - When reporting p values please report as p<0.001 and where higher as the exact p value p=0.002, for example. Please refer to Table 3 in the main text. 

Lines 264, 270 - please revise to ‘statistically non-significant’.

Line 268 - given that the difference in prevalence of OASI within group 2A was not statistically significant, it would be better to state the prevalence in 2010 and 2021.

For the aRRs presented, please specify the comparison group.

Discussion:

Please present and organize the Discussion as follows: a short, clear summary of the article's findings; what the study adds to existing research and where and why the results may differ from previous research; strengths and limitations of the study; implications and next steps for research, clinical practice, and/or public policy; one-paragraph conclusion.

Please remove all subheadings within your Discussion e.g. Principal findings. 

Line 350 - please revise to ‘statistically non-significant’

Line 381 - suggest ‘halved’ as an alternative.

Please remove the information on competing interests. In the event of publication, this information will appear in the article metadata, via entries in the submission form.

Tables:

Please provide the unadjusted comparisons as well as the adjusted comparisons. 

It is not clear in the tables how the p values relate to the aRRs provided. Please clarify. For example, it appears from the main text that p values relate to unadjusted comparisons, as the difference in OASI was deemed statistically non-significant after adjustment, yet has a p value in the table p=0.021. 

For the aRRs presented, please specify the comparison group.

References:

Please ensure that journal name abbreviations match those found in the National Center for Biotechnology Information (NCBI) databases (http://www.ncbi.nlm.nih.gov/nlmcatalog/journals), and are appropriately formatted and capitalised. Where website addresses are cited, please specify the date of access. 

Supplementary files: 

Please provide supplementary tables as standalone files. 

Comments from Reviewers:

Reviewer #2: Thank you to the authors for addressing the concerns raised in my previous reviews. I have no further comments.

[LINK]

---

## [Editor Report · Decision Letter 4]

18 Nov 2024

Dear Dr DESPLANCHES, 

On behalf of my colleagues and the Academic Editor, Andrew Shennan, I am pleased to inform you that we have agreed to publish your manuscript "Episiotomies and obstetric anal sphincter injuries following a restrictive episiotomy policy in France: An analysis of the 2010, 2016 and 2021 National Perinatal Surveys" (PMEDICINE-D-24-01340R4) in PLOS Medicine.

PRESS

Sincerely, 

Rebecca Kirk

On behalf of:

Louise Gaynor-Brook, MBBS PhD 

Senior Editor 

PLOS Medicine